# Atomistic simulations indicate the c-subunit ring of the $F_1F_o$ ATP synthase is not the mitochondrial permeability transition pore

Wenchang Zhou, Fabrizio Marinelli*, Corrine Nief, José D Faraldo-Gómez*

Theoretical Molecular Biophysics Section, National Heart, Lung and Blood Institute, National Institutes of Health, Bethesda, United States

**Abstract** Pathological metabolic conditions such as ischemia induce the rupture of the mitochondrial envelope and the release of pro-apoptotic proteins, leading to cell death. At the onset of this process, the inner mitochondrial membrane becomes depolarized and permeable to osmolytes, proposedly due to the opening of a non-selective protein channel of unknown molecular identity. A recent study purports that this channel, referred to as Mitochondrial Permeability Transition Pore (MPTP), is formed within the c-subunit ring of the ATP synthase, upon its dissociation from the catalytic domain of the enzyme. Here, we examine this claim for two c-rings of different lumen width, through calculations of their ion conductance and selectivity based on all-atom molecular dynamics simulations. We also quantify the likelihood that the lumen of these c-rings is in a hydrated, potentially conducting state rather than empty or blocked by lipid molecules. These calculations demonstrate that the structure and biophysical properties of a correctly assembled c-ring are inconsistent with those attributed to the MPTP.

**\*For correspondence:** fabrizio. marinelli@nih.gov (FM); jose. faraldo@nih.gov (JDF-G)

**Competing interests:** The authors declare that no competing interests exist.

## Introduction

Under certain pathological conditions mitochondria cease ATP production and instead trigger apoptosis. At the onset of this process is the depolarization of the inner mitochondrial membrane, which under normal conditions sustains the proton-motive-force powering the ATP synthase. It has long been hypothesized that this depolarization is mediated by a channel protein referred to as the Mitochondrial Permeability Transition Pore (MPTP) (*Szabó and Zoratti, 1992*; *Bernardi and Forte, 2007*; *Halestrap, 2009*; *Azzolin et al., 2010*). The hallmark properties of this putative channel are: very large ionic conductance (up to 1300 pS); high permeability to small molecules, including osmolytes (less than 1.5 kDa); and no solute specificity (aside from size-limit) or ion selectivity. Thus, opening of the MPTP, thought to be induced, for example, by excessive amounts of $Ca^{2+}$ or ROS in the mitochondrial matrix, is catastrophic for the cell: not only does it short-circuit the respiratory chain, causing the ATP synthase to reverse its activity and deplete cellular ATP, but it also induces the osmotic swelling of the matrix, rupturing the outer membrane and causing the release of pro-apoptotic proteins (*Halestrap, 2009*).

The molecular identity of the MPTP remains to be established. Seemingly promising studies have implicated mitochondrial proteins as diverse as the outer-membrane Voltage-Dependent Anion Channel (VDAC), the inner-membrane Adenine Nucleotide Translocase (ANT), and the non-transmembrane cyclophilin D; ultimately, however, none of these proteins was found to be actually essential (*Bernardi and Forte, 2007*; *Halestrap, 2009*; *Azzolin et al., 2010*; *Bernardi et al., 2015*). More recently, Jonas and co-workers have proposed that a sub-complex of the mitochondrial $F_1F_o$ ATP

synthase, known as the c-ring, is the MPTP (*Alavian et al., 2014*). The c-ring is an oligomer of identical copies of subunit-c, each of which is a hairpin of two transmembrane helices (*Figure 1—figure supplement 1A*); when assembled, these helices are arranged in two concentric rings around a central lumen (*Meier et al., 2005*; *Pogoryelov et al., 2009*; *Symersky et al., 2012*). The $F_1$ domain, which projects out of the membrane and contains the catalytic domain, sits atop the ring (*Figure 1—figure supplement 1B*) (*Dautant et al., 2010*; *Watt et al., 2010*). In the operating enzyme, the membrane $F_o$ domain works like a turbine, with the c-ring rotating against other static elements as protons sequentially bind to each of the c-subunits and are released across the membrane after a revolution (*Meier et al., 2011*; *Leone and Faraldo-Gómez, 2016*). These proton-binding sites are on the outer surface of the ring, and feature a conserved glutamate and other polar groups (*Figure 1—figure supplement 1A*) (*Pogoryelov et al., 2009*; *Symersky et al., 2012*; *Vollmar et al., 2009*; *Zhou et al., 2016*); the lumen, by contrast, is largely hydrophobic. What Jonas et al. propose is that the metabolic conditions associated with the mitochondrial permeability transition induce the $F_1$ and $F_o$ sectors to dissociate, upon which the lumen of the c-ring would become the MPTP (*Alavian et al., 2014*).

Here, we use all-atom molecular dynamics simulations to directly investigate the plausibility of this claim, which has been questioned by others (*Bernardi et al., 2015*; *Halestrap, 2014*). Specifically, we consider two c-rings of different size, and examine their ion-conducting properties under the assumption that their lumen is hydrated. Independently, we examine the validity of this assumption, that is, we evaluate whether a hydrated lumen, which is a requisite for ion/solute permeation, is more or less probable than an empty lumen or a lumen occupied by a lipid bilayer, which would not permit conduction at the expected rates. Taken together, our results strongly suggest that the c-subunit ring, if correctly folded and assembled, is not the elusive MPTP, even if it did in fact became detached from the $F_1$ sector.

## Results and discussion

### The hydrated c-ring lumen does not conduct like the MPTP

We first considered the $c_{10}$-ring from *Saccharomyces cerevisiae*, which is the only mitochondrial c-ring of known atomic structure (*Figure 1A*) (*Symersky et al., 2012*; *Zhou et al., 2016*). The lumen of this ring is also wider than that of the $c_8$-rings found in vertebrates (*Figure 1—figure supplement 2A*) (*Watt et al., 2010*), and thus, it is in principle more likely to sustain a larger conductance. (Note the mitochondrial permeability transition has also been observed in yeast (*Azzolin et al., 2010*)). To evaluate the conducting properties of this $c_{10}$-ring, we calculated the free-energy profile associated with the permeation of either $K^+$ or $Cl^-$ across the lumen, as well as the diffusion-coefficient profile for each ion (*Figure 1B–D*), in a state in which the c-ring lumen is filled with water at bulk-like density (as will be discussed below, this state is intrinsically metastable though not the most probable). The resulting conductance values are 2.5 ps for $K^+$ and 116 ps for $Cl^-$ (for 100 mM KCl). These values are clearly inconsistent with the properties of the MPTP, not only in terms of their magnitude, but also in that they reveal a marked anion selectivity. Analogous calculations carried out in the absence of electrolytes indicate that this selectivity owes to a ring of arginine side-chains at the mouth of the lumen, on the matrix side, which influences the energetics and dynamics of ion permeation electrostatically (*Figure 1G and A*). This electropositive arginine ring, which is conserved in other mitochondrial c-rings (*Figure 1—figure supplement 2A*), counters the cost of dehydration for $Cl^{-}$, but imposes an additional barrier for $K^+$ (*Figure 1C*), incompatible with the high conductance properties of the MPTP.

### The lumen of the mitochondrial c-ring is not hydrated

Because the interior surface of the c-ring is highly hydrophobic, the permeation of ions and osmolytes across the lumen requires an aqueous pathway. Thus, another way to evaluate the notion that the c-ring is the MPTP is to determine whether this lumen is indeed likely to be filled with water. We therefore sought to calculate the free-energy gain or cost associated with varying the density of water inside this region. To do so, we developed a variation of the Metadynamics enhanced-sampling technique, with which we simulated multiple transitions between an empty state of the c-ring lumen and another that is maximally hydrated, while quantifying the energetics of this process

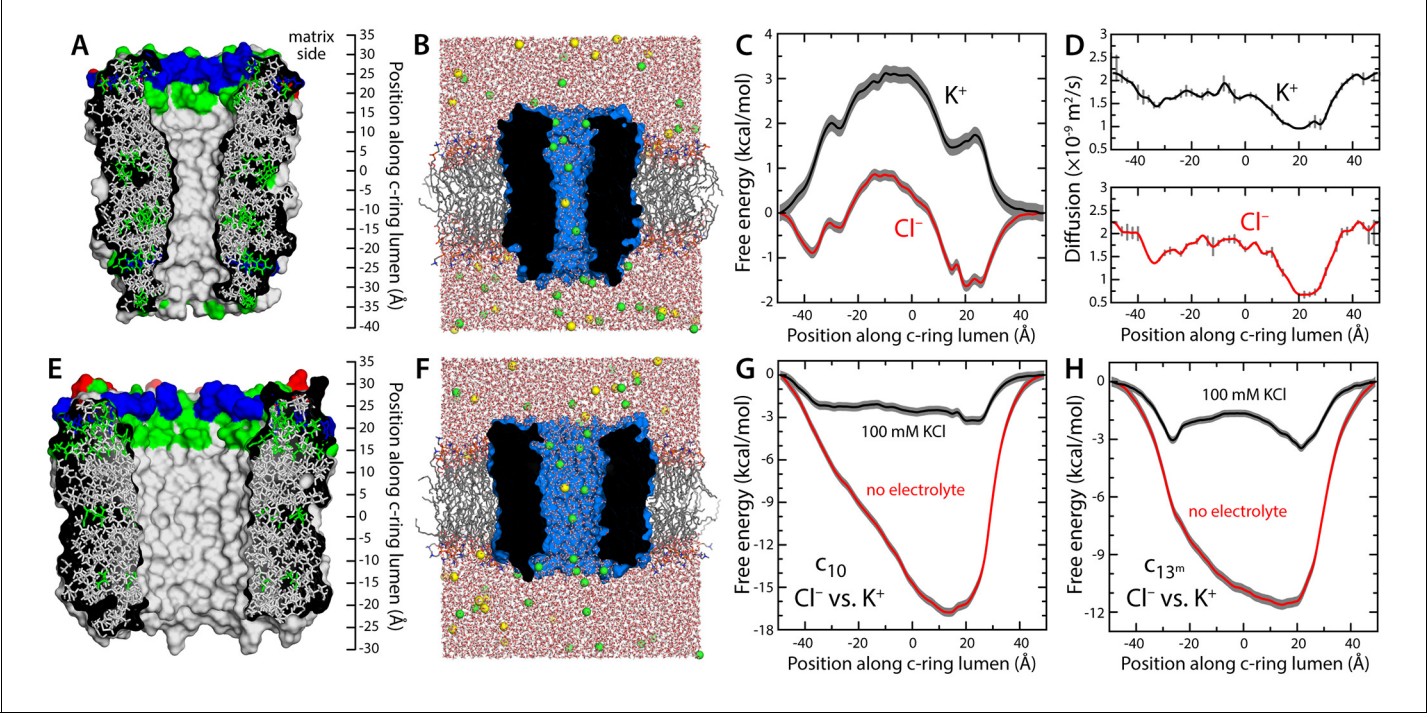

**Figure 1.** Evaluation of the ion-conducting properties of the c-ring lumen, assuming a hydrated state. (A) Cross-section of the $c_{10}$-ring of the mitochondrial ATP synthase from *S. cerevisiae*. The surface of the protein is colored as follows: Lys and Arg, blue; Asp and Glu, red; other polar residues (and protonated Glu), green; other residues, grey. The ring is oriented such that the interface with the $F_1$ domain, inside the mitochondrial matrix, is up. (B) Molecular simulation system employed to study the properties of the $c_{10}$-ring, shown again in cross-section (blue). The ring is embedded in a model phospholipid bilayer, in 100 mM KCl. $K^+$ and $Cl^-$ ions are shown as yellow and green spheres, respectively. Note the c-ring lumen is hydrated. (C, D) Potential-of-mean-force ($G(z)$, PMF) and diffusion-coefficient profiles for the permeation of either $K^+$ or $Cl^-$ across the lumen of the $c_{10}$-ring (Materials and methods). The lack of binding sites for $K^+$ and the 3 kcal/mol free-energy barrier explain the modest $K^+$ conductance; permeation by $Cl^-$, by contrast, is strongly favored electrostatically. (E, F) Same as (A, B), for a variant of the $c_{13}$-ring from *Bacillus pseudofirmus* (*Figure 1—figure supplement 2B,C*), whose lumen is wider than that of the $c_{10}$-ring. (G) Free-energy of selectivity for $Cl^-$ and against $K^+$ by the $c_{10}$-ring lumen, examined with and without electrolyte. The selectivity profile, $\Delta G(z)$, was calculated by subtracting the individual PMF profiles, $G(z)$, in each case. The marked increase in $Cl^-$ selectivity toward the lumen entrance, on the matrix side, confirms the strong electrostatic influence of a ring of arginine residues. (H) Same as (G), for the variant of the $c_{13}$-ring from *B. pseudofirmus*. Despite its wider lumen, this ring is also markedly anion selective, unlike the MPTP.

The following figure supplements are available for figure 1:

**Figure supplement 1.** Structure of the $F_1c_{10}$-ring subcomplex of the yeast mitochondrial ATP synthase.

**Figure supplement 2.** Comparison of c-rings.

(Materials and methods). The resulting free-energy profile (*Figure 2A*) reveals that a metastable state does exist in which the lumen of the mitochondrial c-ring is filled with water, at bulk-like density (*Figure 2B*). However, this water-filled state is approximately 15 kcal/mol less favorable than a second state in which most of the lumen of the c-ring is void of any water (*Figure 2B*). That is, the probability that the c-ring lumen exists in a 'conducting state' is negligible – a dehydrated, non-conducting state is much more probable, by multiple orders of magnitude.

## The lumen of the mitochondrial c-ring is plugged by lipid molecules

Irrespective of the question of hydration, the notion that the c-ring is the MPTP implicitly assumes that lipid molecules are somehow excluded from its lumen as c-subunits gradually assemble to form a ring. There is, however, no evidence that this is the case; to the contrary, detergent molecules are often observed in the lumen in X-ray structures of c-rings (*Pogoryelov et al., 2010*; *Schulz et al.,*

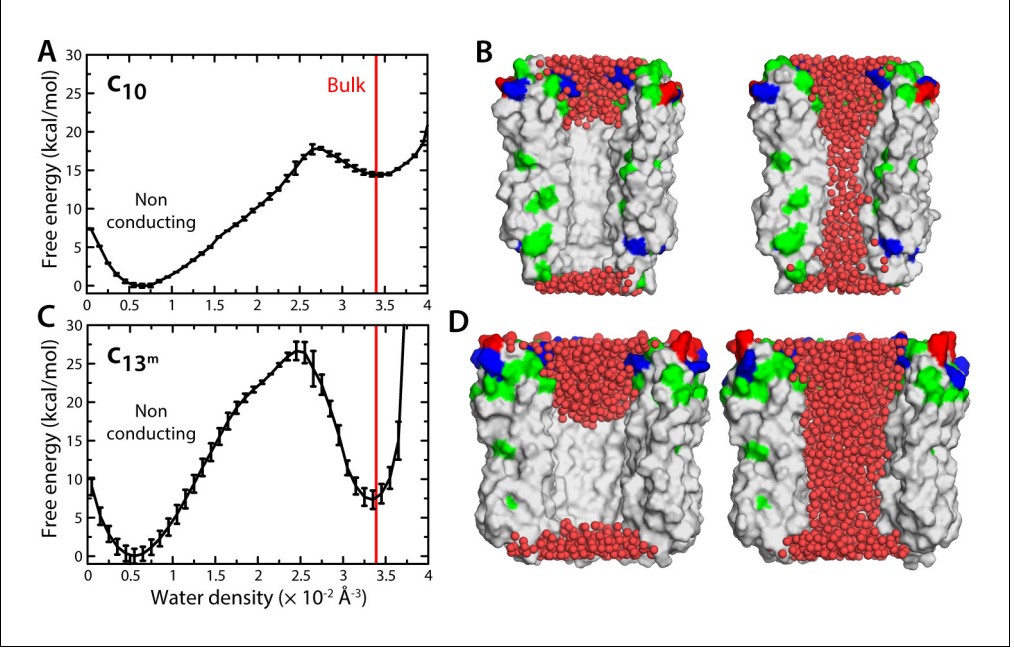

**Figure 2.** Evaluation of the likelihood of hydration of the c-ring lumen. (**A**) Free energy as a function of the water density inside the lumen of the mitochondrial $c_{10}$-ring, calculated with a variant of the Metadynamics technique (Materials and methods), for a simulation system analogous to that shown in *Figure 1B*. The density value for bulk water (for the CHARMM36 forcefield) is indicated in red. Error bars reflect the differences between two profiles calculated using different halves of all simulation data. (**B**) Snapshots of the c-ring lumen in the two metastable minima detected in the free-energy profile shown in panel (**A**), i.e. a non-conducting, de-wetted state (left), and a water-filled, putatively-conducting state (right), whose properties are characterized in *Figure 1*. For clarity, several c-subunits in the c-ring are omitted, as are the lipid bilayer and the solvent outside the lumen. Hydrogen atoms in water (red spheres) are also omitted. Note that in the non-conducting state (left), the two regions at the entrance of the hydrophobic portion of the lumen are hydrated; hence, the density value for this state in free-energy profile in panel (**A**) is not zero. (**C**, **D**) Same as (**A**, **B**), for the variant of the prokaryotic $c_{13}$-ring from the *B. pseudofirmus* ATP synthase.

The following figure supplement is available for figure 2:

**Figure supplement 1.** Graphical definition of the density collective-variable with cylindrical geometry.

*2013*; *Matthies et al., 2014*), and AFM images of two-dimensional ring arrays also indicate the lumen might be occluded (*Meier et al., 2001*). To conclusively evaluate this question, we carried out three independent simulations in which the mitochondrial c-ring lacked two of the c-subunits (*Figure 3*). This open c-ring resembles the kind of assembly intermediate that has been observed in AFM micrographs of prokaryotic c-rings (*Müller et al., 2001*; *Pogoryelov et al., 2005*). Initially, we assumed the lumen of the c-ring to be hydrated (*Figure 2B*), as this is the necessary feature of the hypothesized conducting form. As shown in *Figure 3A–C*, however, the three simulations consistently demonstrate that all this water is quickly displaced by lipid molecules gradually entering the c-ring lumen; ultimately, these lipid molecules occupy the totality of the pore, in a bilayer-like arrangement akin to that outside the c-ring (*Figure 3C*). There is no reason to assume that the incorporation of the two missing c-subunits would alter the lipid occupancy of the lumen, which would therefore remain in a non-conducting state when the c-ring assembles completely.

## Enlargement of c-ring lumen does not explain properties of MPTP

One of the elements in the proposal that the mitochondrial c-ring is the MPTP is that the size of the lumen might expand upon dissociation of the $F_1$ sector (*Alavian et al., 2014*). This expansion is not entirely implausible; indeed, subtle variations in the amino-acid sequence of the c-subunits can

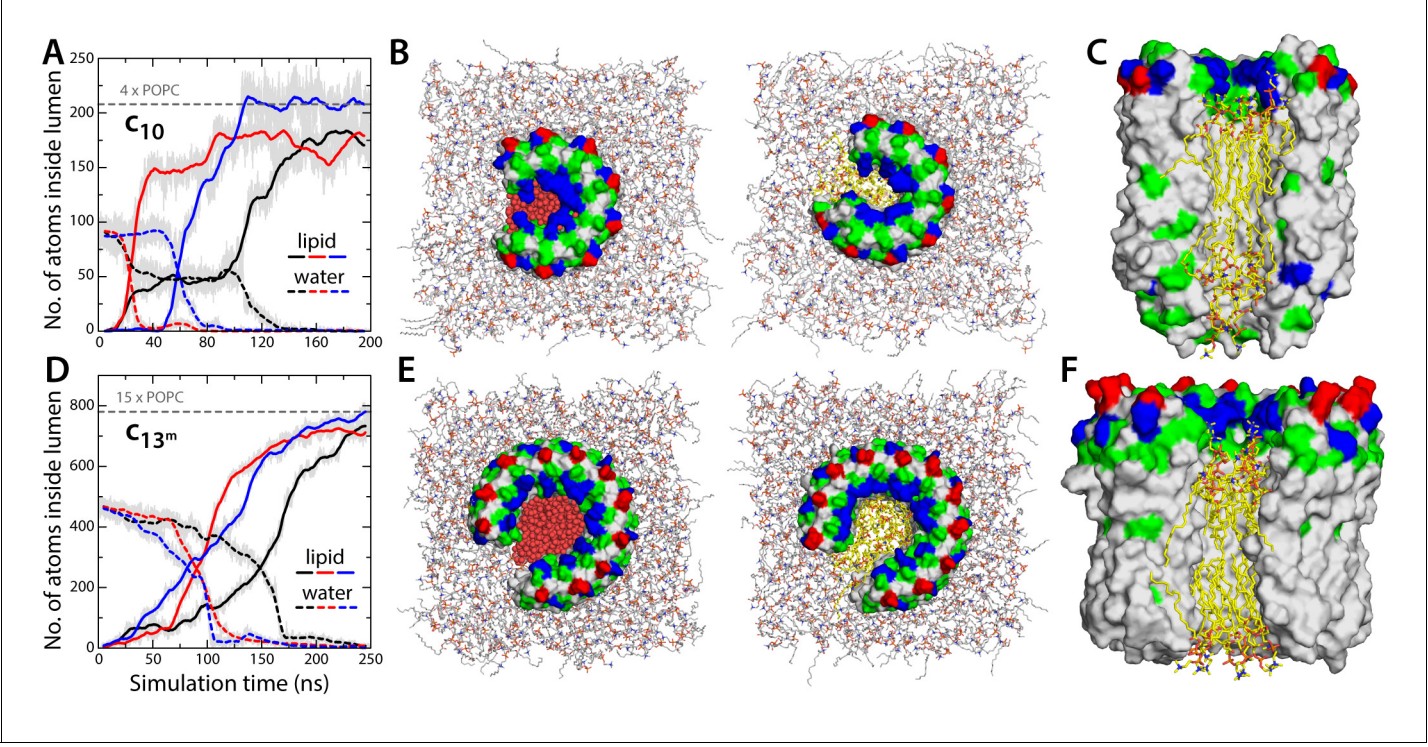

**Figure 3.** Evidence that lipid molecules block the lumen of the c-ring. Three independent molecular dynamics simulations were carried out of assembly intermediates of the $c_{10}$- and $c_{13}$-rings lacking two of the c-subunits. Initially, the lumen of these c-rings was hydrated at bulk-like density. (A) Time-series of the number of non-hydrogen atoms from either water (dashed lines) or lipid molecules (solid lines) inside the lumen of the mitochondrial c-ring. The volume consider for this count is a cylinder of radius 7 Å and height 32 Å, approximately centered in the middle of the membrane. Running averages are shown for each of the three simulation, colored in black, red and blue, respectively, with the raw data shown in the background in grey. The number of atoms equivalent to 4 POPC lipid molecules is indicated, for reference. (B) Snapshots of the molecular system, at the beginning (left) and at the end (right) of the simulation. Water molecules (red) are progressively displaced by lipid molecules (shown in yellow) entering the lumen laterally. (C) Side-view of the open c-ring at the end of the one of the simulations. The lipid molecules inside the lumen preserve a bilayer arrangement, and adapt to the specific features of the protein surface. (D, E, F) Same as (A, B, C), for the variant of the $c_{13}$-ring from *B. pseudofirmus*. The volume considered in panel (D) is a cylinder of radius 14 Å and height 34 Å.

destabilize the native state of a c-ring, and favor the assembly of larger c-rings with a slightly greater number of c-subunits (*Pogoryelov et al., 2012*). With this in mind, we repeated all calculations presented thus far for a variant of the $c_{13}$-ring from the bacterium *Bacillus pseudofirmus* (*Figure 1E–F*). Among the ATP synthase c-rings of known structure, this specific $c_{13}$-ring features the widest lumen (*Preiss et al., 2010*). Thus, although no process akin to the permeability transition has been described in bacteria, we reasoned this ring would be a suitable system to assess whether an expanded mitochondrial c-ring might be the MPTP. For a better comparison, two of the residues lining the lumen of the ring were mutated, to make it more similar to the mitochondrial c-ring (*Figure 1—figure supplement 2B,C*). The results from this new set of calculations, however, confirm the conclusions drawn above. Because the lumen is wider, the permeability of the hydrated state is significantly larger, but this state remains highly anion-selective (*Figure 1H*), unlike the MPTP. Moreover, this water-filled state is still disfavored, relative to the empty lumen, albeit to a lesser degree than in the $c_{10}$-ring (*Figure 2C,D*). Importantly, though, independent simulations of an assembly intermediate lacking two c-subunits demonstrate that the preferred state is again one in which lipid molecules occlude the lumen, in a bilayer arrangement (*Figure 3D–F*).

## Conclusions

This study demonstrates that the characteristics of the lumen of the mitochondrial c-ring are incompatible with the conducting properties of the MPTP. Consistent with the fact that the $F_1$ domain

does not actually seal the entrance to the lumen of the c-ring in the fully assembled, operating enzyme (*Figure 1—figure supplement 1C*), the interior of the ring is not water-filled, but occupied by lipid molecules; that is, it is in a non-conducting state, thereby precluding the dissipation of the ion-motive-force that powers ATP synthesis. Our results also show that even if the c-ring became dissociated from the $F_1$ domain (*Alavian et al., 2014*), despite the nanomolar affinity of this interaction (*Pogoryelov et al., 2008*), and even if its lumen was to become hydrated, under some hypothetical in vitro conditions, the conductance levels sustained by the c-ring would be unlike those of the MPTP.

Admittedly, this study does not rule out that an as-yet-unknown molecular structure consisting of c-subunits, distinct from that of the c-ring, matches the properties of the MPTP. However, it is also worth noting that no evidence of such alternative structure exists, to our knowledge. Structural studies of the c-subunit and its oligomers date back two decades; these studies include two- and three-dimensional crystallography, liquid and solid-state NMR spectroscopy, atomic-force microscopy, and more recently, single-particle cryo-electron microscopy (*Meier et al., 2011*; *Leone and Faraldo-Gómez, 2016*; *von Ballmoos et al., 2009*). Across a wide variety of species and experimental conditions, the consistent conclusion from these studies is that the c-subunits assemble in ring-like oligomers such as those analyzed here. This conclusion holds true for wild-type, mutagenized and inhibited forms of the c-ring, and for a variety of expression systems (including cell-free). Moreover, to our knowledge no apparent discrepancy exists between this set of structural data and the even larger collection of biochemical and functional measurements gathered for this enzyme or its constituent elements. Indeed, in the instances where clear thermodynamic data has been attained, for example, on $Na^+/H^+$ binding to isolated c-rings, it was found to be not only quantitatively consistent with kinetic measurements for the complete enzyme functioning in membranes, but also explicable based on existing structures and theoretical considerations (*Murata et al., 2008*; *Leone et al., 2015*). In absence of concrete experimental evidence of the existence of an alternative architecture, the logical conclusion from this body of work and the present study is therefore that the c-ring from the mitochondrial ATP synthase is not the MPTP.

## Materials and methods

### Simulation systems and general specifications

The molecular dynamics (MD) simulation systems used to study the mitochondrial $c_{10}$-ring from *S. cerevisiae* were adapted from a system used in earlier studies of the proton-binding sites in this structure (*Symersky et al., 2012*; *Zhou et al., 2016*). Briefly, the $c_{10}$-ring had been embedded in a hydrated (and pre-equilibrated) palmitoyl-oleoyl-phosphatydyl-choline (POPC) lipid bilayer, using GRIFFIN (*Staritzbichler et al., 2011*), and equilibrated extensively through a series of restrained and unrestrained MD simulations. Here, the four lipid molecules occupying the center of the c-ring were removed, and instead the lumen was initially filled with water molecules. Several alternative systems were prepared: for the complete ring, one system included 100 mM KCl, while the other included counter-ions only; a simulation system was also prepared in which the c-ring lacks two c-subunits, thus resembling an assembly intermediate. After making these changes, energy minimizations, restrained and unrestrained molecular dynamics simulations were used to equilibrate these systems.

The simulation systems for the $c_{13}$-ring from *B. pseudofirmus* OF4 were also adapted from a system used in a previous study of its proton-binding sites (*Leone et al., 2010*). The central lumen of this ring is about two-fold wider than that of the mitochondrial $c_{10}$ ring, but the amino-acid composition of the protein surface lining this lumen differs. Thus, to make the $c_{13}$-ring more similar, we introduced two mutations (K26L and E30Q) at the entrance of the lumen, on the side that would be exposed to the mitochondrial interior or matrix. In addition, the POPC lipid molecules inside the central lumen were also replaced by water. The resulting systems (with and without added 100 mM KCl, for the complete $c_{13}$-ring, plus a system missing two c-subunits) were equilibrated analogously to the $c_{10}$-ring systems.

All MD simulations were performed with NAMD2 (*Phillips et al., 2005*), using the CHARMM36 forcefield (*MacKerell et al., 1998*; *Best et al., 2012*; *Klauda et al., 2010*), at constant temperature (298 K) and pressure (1 atm) and with periodic boundary conditions in all directions. Long-range electrostatic interactions were calculated using the Particle-Mesh-Ewald algorithm, with a real-space

cut-off of 12 Å. Van der Waals interactions were modeled with a Lennard-Jones potential, cut-off at 12 Å using a smooth switching function taking effect at a distance of 10 Å.

## Ion conductance calculations

Following Hummer and co-workers (*Zhu and Hummer, 2012*), the conductance $\gamma$ of the $c_{10}$-ring lumen was calculated using the expression:

$$\gamma = \frac{q^2 C S}{k_B T} \left[ \int_{Z_1}^{Z_2} \exp\left\{ \frac{G(z)}{k_B T} \right\} \frac{1}{D(z)} dZ \right]^{-1}$$

where $q$ is the charge of the permeant ion, $C$ is the bulk ion concentration (100 mM), $S$ is the effective cross-section area of the lumen (201 Å² for $c_{10}$-ring and 616 Å² for $c_{13}$-ring), $k_B$ is the Boltzmann constant and $T$ the temperature. $G(z)$ is the one-dimensional free-energy landscape reflecting the translocation of the ion across the protein, and $D(z)$ is the position-dependent diffusion coefficient of the ion along the lumen; both $G(z)$ and $D(z)$ were calculated using MD simulations, as described below. It is important to note that this computational framework has been shown to produce conductance values that are in good agreement with experimental measurements (*Zhu and Hummer, 2012*).

## Free-energy profiles for K⁺ and Cl⁻ permeation

The Adaptive-Biasing-Force (ABF) method (*Hénin et al., 2010*; *Fiorin et al., 2013*) was used to calculate the potential-of-mean-force associated with the translocation of either K⁺ and Cl⁺ along the lumen of the c-ring, or $G(z)$, for each of the simulation systems described above. The lumen of the c-ring was filled with water (at bulk-like density) throughout these calculations. (It was not required to impose this condition, as this state is metastable and sufficiently long-lived to observe tens of permeation events). The reaction coordinate was the Z-coordinate of one K⁺ or one Cl⁻ ion, defined relatively to the Z-coordinate of the center-of-mass of a ring of Cα atoms in the protein (from residues A22 in the $c_{10}$-ring, and from A18 in the $c_{13}$-ring). The permeating ion was confined to a cylindrical volume (through flat-bottom restraining potentials) whose center and axis coincide with the center and axis of the lumen (defined by two additional rings of Cα atoms), and whose radius is slightly larger, namely 8 Å for the $c_{10}$-ring, and 14 Å for the $c_{13}$-ring. The length of the cylinder along the Z-coordinate was 100 Å, therefore projecting into the bulk solution on both sides of the membrane.

For each ion type and system, two ABF simulations of 300 ns each were carried out, with the permeant ion starting on opposite sides of the membrane; in the course of each of these trajectories, the ions traverse the lumen multiple times, in both directions. The range in the Z-coordinate to be sampled was divided up in 0.2 Å bins, and 1000 unbiased samples were collected for each bin before the estimated biasing force was applied and updated. A third simulation was then conducted similarly for 100 ns to combine and refine the bias potentials calculated from the first two simulations. The free-energy profiles shown in *Figure 1* reflect the outcome of these final calculations, while the error bands reflect the differences between the two initial profiles (averaged over the length of the profile).

## Diffusion profiles for K⁺ and Cl⁻ across the lumen

To calculate the diffusion coefficient of the permeant ion as a function of its position along the c-ring lumen, or $D(z)$, 51 independent simulations of 10 ns each were carried out for each system. In each of these simulations, the permeant ion was confined (through flat-bottom restraining potentials) in a disk of 2 Å in length (and radius as specified above), whose center varies in the Z-direction from −50 to 50 Å (relative to the protein, as described above). The starting configurations for each of these simulations were obtained from the ABF trajectories. The position-dependent diffusion coefficient was calculated *a posteriori* from the expression (*Zhu and Hummer, 2012*):

$$D(z) = \frac{var(z)}{\tau(z)}$$

where the numerator is the variance of the $Z$-coordinate of the ion (relative to the protein), i.e. $var(z) = \langle z^2 \rangle - \langle z \rangle^2$, and $\tau(z)$ is the characteristic time of its autocorrelation function, i.e.:

$$\tau(z) = \int\limits_0^\infty \langle \delta z(0)\, \delta z(t) \rangle \, dt \, / \, var(z)$$

where $\delta z(t) = z(t) - \langle z \rangle$, and $t$ is the so-called lag time. In our simulations, the values of $\tau(z)$ are all well converged for $t > 80$–90 ps. The diffusion profiles shown in *Figure 1* are (cubic-spline) interpolations of the 51 values of $D(z)$ obtained using this approach; the error bars reflect the differences between values obtained from different halves of the simulations.

## Density-biased multiple-walker metadynamics

The Metadynamics method was used to induce the wetting and de-wetting of the lumen of the $c_{10}$ and $c_{13}$-rings reversibly, and to calculate the associated free-energy changes. The collective variable biased in these simulations is the water density in the volume of the lumen. Following Feig and co-workers, we express this density as (*Mirjalili and Feig, 2015*):

$$\rho = \frac{\sum_i \xi(X_i)}{V}$$

where $0 \leq \xi(X_i) \leq 1$ is a weight function of the Cartesian coordinates of a given atom $i$ at a given time, whose value depends on whether that atom is inside ($\xi(X_i) \sim 1$) or outside ($\xi(X_i) \sim 0$) the volume $V$ considered. Given the geometry of the lumen of the c-rings, we used a weight function $\xi(X)$ that defines a cylinder of radius $R$, whose axis and center coincide with the those of the lumen, and which extends from $-Z$ to $+Z$. Importantly, the boundaries of this cylinder are not abrupt, but rather smoothened over an interval $\Delta R$ and $\Delta Z$ around $R$ and $\pm Z$ (*Figure 2—figure supplement 1A*). More specifically:

$$\xi(X) = \xi_z(z)\,\xi_R(r) = \begin{cases} 1 & \text{if } r < R - \Delta R/2 \text{ and } |z| < Z - \Delta Z/2 \\ 0 & \text{if } r > R + \Delta R/2 \text{ or } |z| > Z + \Delta Z/2 \\ \chi_Z(z)\,\chi_R(r) & \text{otherwise} \end{cases}$$

where $\chi_S(\sigma)$ ($\sigma$ denotes $z$ or $r$ and $S$ denotes $R$ or $Z$) is a switching function whose value varies smoothly from 1 to 0 in the intervals $S - \Delta S/2 \leq \sigma \leq S + \Delta S/2$, constructed so that its derivative is 0 at the outer boundaries of the cylinder (*Figure 2—figure supplement 1B*):

$$\chi_S(\sigma) = \frac{\left[(S+\Delta S/2)^2 - \sigma^2\right]^2 \left[(S+\Delta S/2)^2 + 2\sigma^2 - 3(S-\Delta S/2)^2\right]}{\left[(S+\Delta S/2)^2 - (S-\Delta S/2)^2\right]^3}$$

Lastly, the effective volume $V$ of this cylinder is:

$$V = \int \xi(X)\, dX = \int\limits_{-(Z+\Delta Z/2)}^{Z+\Delta Z/2} \xi_Z(z)\, dz \int\limits_0^{R+\Delta R/2} 2\pi r\, \xi_R(r)\, dr$$

As in standard Metadynamics, a time-dependent biasing potential $U_b$ acting on the collective-coordinate $\rho$ was adaptively constructed throughout the simulation; at convergence, the free-energy profile is the negative of this potential. At a given time, the forces acting on a given atom $i$ due to this biasing potential can be derived using the chain rule, that is:

$$\boldsymbol{F}_i^b = -\nabla U_b(\rho) = -\frac{dU_b(\rho)}{d\rho}\left(\frac{\partial \rho}{\partial x_i}, \frac{\partial \rho}{\partial y_i}, \frac{\partial \rho}{\partial z_i}\right)$$

where $x_i$, $y_i$ and $z_i$ are the coordinates of atom $i$. Note that our definition of the weight function $\xi(X)$ is such that $\boldsymbol{F}_i^b$ is non-zero only if atom $i$ is found within the smooth boundaries of the cylinder, that is, in the region where the switching functions $\chi_Z(z_i)$ and $\chi_R(r_i)$ are non-zero.

The collective variable $\rho$ defined above is a function of all the atoms considered in the density calculation. In our case, these atoms are all the water oxygen atoms in the simulation system, which in principle would make the calculation of the density collective variable and related biasing forces prohibitively slow. However, because at a given time the value of the weight function (and associated forces) is non-zero only for the atoms found inside the target volume, we can speed up the calculations by introducing a 'neighbor search' in the algorithm, whereby a list of water molecules within and in the vicinity of that volume is created and updated regularly, but not at every simulation step (*Figure 2—figure supplement 1A*). In addition, we note that unlike earlier work (*Mirjalili and Feig, 2015*), our implementation defines $\xi(X)$ in reference to the protein coordinates, that is, not in absolute Cartesian space, and therefore the tumbling and diffusion of the ring in the membrane is unrestricted during the simulations.

The geometric definitions of the target volumes considered for quantification of the free-energies of hydration of the $c_{10}$ and $c_{13}$-ring lumens are indicated in *Figure 2—figure supplement 1C,D*. The Metadynamics simulations were carried out using eight concurrent replicas, or walkers, which update and share a common bias potential but sample different configurations. The biasing potential applied to the density variable $\rho$ consisted of a series of Gaussian functions of width 0.0005 $Å^{-3}$, added in 4-ps intervals. The height of the Gaussians was gradually diminished throughout the simulations, ultimately reaching a value of 0.0035 kcal/mol; the free-energy profiles were obtained by time-averaging the biasing potential after this time-point (*Marinelli et al., 2009*). The total simulation times per walker were 140 ns for the $c_{10}$-ring (using the last 50 ns for analysis), and 120 ns for the $c_{13}$-ring (using the last 80 ns for analysis). The 'neighbor search' comprised a region that is 4 Å larger than the target volume in all directions, and the list of atoms therein was updated every 200 simulation steps, resulting in a ~ 1000-fold speed-up of the calculations.

## Acknowledgements

This work was funded by the Division of Intramural Research of the National Heart, Lung and Blood Institute, National Institutes of Health.

## Additional information

### Funding

| Funder | Author |
| --- | --- |
| National Heart, Lung, and Blood Institute | Wenchang Zhou<br>Fabrizio Marinelli<br>José D Faraldo-Gómez |

The funders had no role in study design, data collection and interpretation, or the decision to submit the work for publication.

### Author contributions

WZ, Formal analysis, Investigation, Writing—original draft, Writing—review and editing; FM, Software, Formal analysis, Investigation, Methodology, Writing—original draft, Writing—review and editing; CN, Formal analysis, Investigation, Writing—original draft; JDF-G, Conceptualization, Formal analysis, Supervision, Investigation, Writing—original draft, Writing—review and editing

### Author ORCIDs

Wenchang Zhou, http://orcid.org/0000-0003-0397-1032
José D Faraldo-Gómez, http://orcid.org/0000-0001-7224-7676

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
