## [Decision Letter]

Thank you for submitting your article "The c-Subunit Ring of the F1Fo ATP Synthase is Not the Mitochondrial Permeability Transition Pore" for consideration by *eLife*. Your article has been favorably evaluated by Richard Aldrich (Senior Editor) and two reviewers, one of whom, Yibing Shan, is a member of our Board of Reviewing Editors.

The reviewers have discussed the reviews with one another and the Reviewing Editor has drafted this decision to help you prepare a revised submission.

Summary:

Zhou el at. used fully atomistic molecular dynamics simulations to characterize the c-subunit ring of F1F0 ATP synthases as a membrane pore. In this sophisticated and well executed simulation study, the conductance of the channel was shown to be low and selective, inconsistent with the properties of mitochondrial permeability transition pore (MPTP), which plays a crucial role in rupture of mitochondrial envelope in cell death. Moreover, the lumen of the ring was shown to be likely dehydrated, suggesting that the apparent pore is non-conductive at its ground state. The authors thus concluded that the MPTP cannot be the lumen of the c-subunit ring of F1F0 ATP synthases. This work is not the first in questioning the notion of the c-subunit ring as the MPTP, but it has added to the debate crucial and otherwise unavailable atomic details using MD simulations.

Essential revisions:

The reviewers agree with one another in their primary concern to this work, that is that it does not seem to rule out the possibility of another structural configuration of the c-subunit ring different from the one that has been simulated. It is perceivable that in the alternative configuration the c-subunit ring be more hydrated, more conducive, and less selective, and thus more MPTP-like. This concern is accentuated by the report (Alavian et al., 2014, PNAS) of the existence of an open and a closed state of the c-subunit ring, where the closed state is conductive at ~100-300 pS (Figure 1 of the paper), which agrees very well with the simulation conductance, although the ion selectivity is different. The authors did try to study the conductance change upon widening of the pore. But apart from widening of the pore, there might be other conformational changes correlated with conductance. There might be a simple and convincing argument to address this concern, but in any case, an in-depth discussion to rule out the alternative (more open) configuration is vitally important to the drawn conclusion.

---

## [Author Response]

*Essential revisions:*

*The reviewers agree with one another in their primary concern to this work, that is that it does not seem to rule out the possibility of another structural configuration of the c-subunit ring different from the one that has been simulated. It is perceivable that in the alternative configuration the c-subunit ring be more hydrated, more conducive, and less selective, and thus more MPTP-like. This concern is accentuated by the report (Alavian et al., 2014, PNAS) of the existence of an open and a closed state of the c-subunit ring, where the closed state is conductive at ~100-300 pS (Figure 1 of the paper), which agrees very well with the simulation conductance, although the ion selectivity is different. The authors did try to study the conductance change upon widening of the pore. But apart from widening of the pore, there might be other conformational changes correlated with conductance. There might be a simple and convincing argument to address this concern, but in any case, an in-depth discussion to rule out the alternative (more open) configuration is vitally important to the drawn conclusion.*

We have added a section entitled “Concluding remarks”, where we discuss the possibility that an alternative structure of a c-subunit assembly is the MPTP. We concur with the reviewers that our study does not specifically rule out this possibility, and admit so in the text. However, we also point out that no evidence of such an alternative structure has been reported, to our knowledge, by any of the many and diverse structural studies of the c-subunit and its oligomers conducted over the past two decades – which, in turn, have consistently helped to explain a large body of biochemical and functional data.